# Quantitative Traits of Interest in Apple Breeding and Their Implications for Selection

**DOI:** 10.3390/plants12040903

**Published:** 2023-02-16

**Authors:** Radu E. Sestras, Adriana F. Sestras

**Affiliations:** 1Department of Horticulture and Landscape, University of Agricultural Sciences and Veterinary Medicine Cluj-Napoca, 3–5 Manastur Street, 400372 Cluj-Napoca, Romania; 2Department of Forestry, University of Agricultural Sciences and Veterinary Medicine Cluj-Napoca, 3–5 Manastur Street, 400372 Cluj-Napoca, Romania

**Keywords:** apple scab, fruit, heritability, offspring, polygenic traits, powdery mildew, quality, selection, yield

## Abstract

Apple breeding is a laborious and long-lasting process that requires qualified resources, land, time, and funds. In this study, more than 5000 F_1_ apple hybrids from direct and testcrosses were analyzed. The results revealed how the phenotypic expression of the main quantitative traits of interest assessed in five half-sib families was controlled by the additive genetic effects and by non-additive effects of dominance and epistasis. The statistical number of hybrids required to ensure efficient selection increased exponentially with the number of desirable traits. The minimum number of progenies required to obtain a hybrid with associated quantitative traits of agronomic interest was highly variable. For two independent traits essential in selection (fruit size and quality), but incorporated together in the same hybrid, the statistical number was between about 30 and 300. If three more cumulative traits were added (a large number of fruits per tree, resistance/tolerance to apple scab, and powdery mildew attack), the limits increased to between 1500 and 18,000. The study highlighted the need for new apple varieties due to the narrowing of the genetic diversity of the cultivated species and how the choice of parents used in hybridizations (as well as the objectives pursued in the selection) can increase the efficiency of apple breeding.

## 1. Introduction

The apple is the primary fruit-bearing species in temperate regions and ranks first in terms of global fruit production. Several factors contributed to the spread of apples over the world, including: the nutritional, gustatory, therapeutic, and prophylactic value of fruits; the specific technological properties and agrobiological characteristics of trees; ecological plasticity; and high economic value [1,2]. Because of apples’ dietary and sanogenic properties as well as their high nutritional value and health benefits, the aphorism ‘An apple a day keeps the doctor away’ has become a truism [3,4]. Apples are a healthy food that contain a wide range of nutrients and non-nutrients. As a result, apples are among the most consumed fruits in many countries and cultures, with a per capita intake of roughly 20–30 kg/year [5,6,7,8]. By 2025 and 2050, it is anticipated that the average daily consumption of fruits will increase from 204 to 242 g per person worldwide [9].

Apples are grown on an area of approximately 4 million ha worldwide, with fruit production averaging around 86–87 million metric tonnes per year in recent years [10]. China is the leading producer of apples worldwide; in the 2019–2020 crop year, apple production amounted to around 41 million tons. Countries with a large apple production countries after China are the United States of America, Turkey, India, and the Russian Federation (these represent the top five countries). Other countries with a large production of apples also include Poland, Italy, France, Germany, Turkey, Argentina, Japan, and Chile. China’s apple production increased rapidly from 4.5 million metric tonnes in 1990 to 40.9 million metric tonnes in 2014, thereby exhibiting a truly remarkable growth in apple acreage and production [10]. Interestingly, a single cultivar (Fuji) is grown on more than 70% of the apple-dedicated area in China [11].

The cultivated apple influenced human history, and the apple is considered a symbol of wisdom and love [12]. Apple varieties have a genetic background with a strong imprint of their ancestors *M. sieversii* and *M. sylvestris* as well as numerous interspecific and intraspecific hybridizations [13]. Hybridization can broaden the genetic basis of cultivars by amplifying the heterozygosity of the offspring, but in general, apple breeding using hybridization was based on the principle of ‘crossing the best with the best parent’. Because the best varieties were also the most well-known and widespread in the world, relatively few parents were used in the new breeding works. Consequently, a small number of cultivars were repeatedly used as parents during the apple breeding process [14,15]. Thus, the most well-known varieties became the parental forms of most new varieties. Therefore, many of the modern varieties found in the world assortment are related to each other and have common parents or close origins or ancestors; i.e., a common genetic basis [16]. Among the most well-known and widespread varieties in the world are Golden Delicious and Red Delicious as well as their mutants [17]. McIntosh, Rome Beauty, Jonathan, Northern Spy, Stayman, York, Cox’s Orange Pippin, Belle de Boskoop, Reinette du Canada, Worcester Pearmain, and James Grieve are other ‘classic’ varieties known and widespread in the world. In recent decades, newer varieties have begun to spread in the culture, including Elstar, Gala, Jonagold, Mutsu, and Pink Lady (all with the Golden Delicious variety as their common parent) or others such as Empire and Fuji (both from Red Delicious), Idared, Granny Smith, Topaz, Pinova, Braeburn, Florina, Arlet, Rubin, Champion, Kanzi, and Jazz [18]. Newly created varieties have given rise to the possibility of higher market pricing even if many traditional types continue to play a vital role in apple culture around the globe. To control the development and marketing of the new cultivars and raise prices for both cultivar owners and producers, ‘variety clubs’ were created. As a result, many fruit growers were limited in their ability to establish new orchards using certain new cultivars [19,20,21,22]. However, even for some more recent cultivars such as Crimson Snow, which has gained attention due to promotion and marketing methods as well as its great commercial look and organoleptic attributes, the ancestry is unknown. Based on SSR research, Crimson Snow, a putative descendant of Cripps Pink, was revealed to be a descendant of Delicious [23].

As with any agricultural species, apple quality and production can vary. The apple can also be impacted by variations in consumer preferences, market demands, the processing industry, farmers’ obstacles and difficulties, and other constraints to sustainable apple production [11,24,25,26]. The appearance of new diseases or pests or an increase in their aggressiveness might have an impact on fruit yield and quality. Oscillations can occasionally be detected not just across longer time periods but also from one year to the next; for example, because of unfavorable weather during particular periods of the year, climatic events, or other unforeseen circumstances [27,28,29]. In 2017, a single night of frost on April 19 decreased apple production in Europe by 24% [30]. In China, severe spring frosts in the northern regions in 2020–2021 had a negative impact on fruit quality and quantity because the frost damaged blossoming trees [31]. Additionally, COVID-19 caused logistical challenges, and the pandemic crisis intensified food insecurity (including the apple sector), causing seasonal labor shortages and issues with supply chains [32]. Instead, it was projected that apple production would increase in some EU countries and Turkey, where relatively good growing conditions and the introduction of new varieties could result in a sixth consecutive year of rising yield [33]. Apple breeding must be prospective and anticipatory in order to mitigate the hazards that could result from the many stresses (abiotic and biotic) or the market pressure and the continuously rising expectations of consumers and producers. Apple breeding goals must be both broad and specific in order to produce results that can be sustained in the long term [34,35,36,37].

Even though remarkable results have been obtained in apple breeding (as evidenced by the multitude of cultivars obtained in various countries around the world), the conservation of genetic resources and the creation of new cultivars are processes that have seen a noticeable slowdown or even stagnation in recent years. The global economic and social situation, the global trend of continuously reducing resources allocated for these purposes, and the risk of biodiversity loss and genetic resource restriction can severely impact the efforts to capitalize on genetic resources and counteract possible risk factors for the cultivated species [38,39]. The apple serves as a model species for cultivated plants (especially for perennial ones) due to its extensive use and socioeconomic significance. To highlight the complexity of apple breeding, a study of the effectiveness of selection in F_1_ hybrid populations obtained from intraspecific hybridizations was conducted. Depending on the manner of parental form participation and the type of cross-breeding performed, the results of numerous hybridizations were compared. Additionally, it was investigated how the parental forms utilized for various hybrid combinations and the criteria employed in apple breeding could impact selection efficiency. The biological material in the field did not benefit from the maintenance activities because of the drastically reduced funding. Consequently, data were collected under these circumstances, in a natural context (i.e., long-term F_1_ hybrid maintenance in the field, absence of tree cutting, fertilization, phytosanitary treatments, etc.).

## 2. Results

The average values of the trunk cross-sectional area (TCSA) and the height of the trees in the 25 hybrid combinations produced by direct crosses between parents with different growth vigor in the F_1_ apple hybrids showed significant differences. A more accurate evaluation of the outcomes was accomplished by dividing the 25 hybrid combinations into three categories based on the vigor of the maternal parents (weak, medium, and strong vigor) (Table 1).

The cross-sectional area of the trunk in F_1_ hybrids on average in the experiment was 54.9 cm^2^, and in hybrid families it ranged between 33.2 cm^2^ (Ardelean × Ancuța) and 71.2 cm^2^ (Ardelean × Prima). The difference between the mean values of the two families with the most extreme TCSA levels resulted in a large range of variation. The average of the hybrids from Ardelean × Prima exceeded twice those of Ardelean × Ancuța, which denoted very strong but completely different influences of Prima and Ancuța as fathers in the transmission of a certain vigor in offspring. The hybrid combinations that produced less vigorous seedlings were: Ancuța × Golden spur, Aromat de vară × Ancuța, Starkrimson × Ancuța, Reinette Baumann × Golden spur, and Roșu de Cluj × Ancuța. Depending on the vigor groups of the maternal parents, the lowest vigor of the F_1_ hybrids was obtained in the maternal parents with low growth vigor. The coefficients of variation (CV%) for TCSA had high values that oscillated between 15.7% (Kinrei × Jonathan) and 71.5% (Ancuța × Roșu de Cluj). When considering the CV% (small (0–10%), medium (10–20%), high (20–30%), and very high (over 30%)), only two combinations had medium variation, four had high, and the rest had very high.

The height of the hybrids showed a large amplitude, and there were noticeable differences amongst the examined combinations, thereby demonstrating the strong parental influences on the trait. The average height of the trees for all hybrid combinations was 3.8 m; the extreme values were found in the Ardelean × Ancuța (3.0 m) and Ardelean × Feleac (5.1 m) combinations. Elite plants with a smaller tree height, in addition to other desired traits, were more likely to occur in the following hybrid populations: Ardelean × Ancuța, Ancuța × Golden spur, Roșu de Cluj × Kaltherer Böhmer, Roșu de Cluj × Golden spur, Kinrei × Jonathan, Roșu de Cluj × Ancuța, Ancuța × Mutsu, Aromat de vară × Ancuța, and Prima × Feleac.

The vigor of the F_1_ hybrids from the five crosses with a common tester revealed obvious differences depending on half-sib families (Table A1) due to the maternal parents because the paternal parent was the same in each combined testcross. Among the HRS selections and varieties used in the hybridization with Feleac as a tester, DSF 7/68 and DSF 3/80 had the highest combining ability for tree vigor assessed as TCSA (the latter for both TCSA and hybrid height). According to current breeding goals, hybrid combinations with the maternal parents X-5-71, X-17-19, and X-3-8 are promising for the selection of new elite plants with low growth vigor. A contrasting situation was found in the Prima cultivar used as the maternal parent. Its progenies were distinguished by a robust appearance and trees with thick trunks but a lower height than the hybrids of all other combinations. TCSA as the mean of hybrid combinations with a Mutsu tester was much lower than with a Feleac tester. The reduced vigor transmitted from Mutsu to their offspring was also confirmed for the F_1_ hybrids’ heights. The testcross with the lowest tester vigor (Starkrimson) was found to have the highest mean TCSA value.

For tree vigor, a total of 5556 F_1_ hybrids originated from direct and testcross hybridizations with parents of different levels of vigor were arbitrarily classified into three classes (Table 2). Overall, the following proportions were obtained for the TCSA: 16.8% weak, 64.7% medium, and 18.6% strong. By dividing the tree height of the F_1_ hybrids into weak, medium, and strong, the proportions obtained in the three categories were 15.8%, 53.1%, and 31.1%.

When the parents were divided into three vigor categories to evaluate the combining ability for all crosses, it was discovered that genetic effects of the general combining ability (GCA) were generally more significant than those of the specific combining ability (SCA) both for TCSA and tree height (Table 3). In accordance with the stated breeding objectives, it is important to create new cultivars with low vigor, and it was confirmed that offspring with low vigor were more frequently produced from parents with low growth vigor. Thus, the parents with weak vigor had the highest positive GCA value for TCSA and tree height. According to the vigor of the parents and how they were combined, additive gene effects played a significant role in the transmission and fixation of vigor in the F_1_ hybrids, albeit in a variety of manners (i.e., in some combinations such as strong × weak vigor parents, negative GCA values were recorded).

The growth and ramification of the trees in some hybrids were rather simply framed in one of the four architectural ‘ideotypes’ established by Lespinasse et al. [40]. Figure 1 depicts the obvious predominance of plants with normal growth and fruiting (84.52%). The spur type accounted for 12.14% of the total, 3.07% of the weeping type, and only 0.27% of the compact-columnar type. The spur proportion was divided between the fastigiate (3.2%) and upright (8.9%) ideotypes and the standard between the upright to spreading (10.2%), spreading (67.1%), and drooping (7.2%) ideotypes using the UPOV classification for edible [41] and ornamental cultivars [42].

In the first two years of their lives, none of the hybrids attained the fruiting period. A relatively small percentage of seedlings (5.7%) produced their first fruit in the third year (Figure 2).

The percentage of hybrids that produced fruit increased gradually in consecutive years and reached 11.7% in the fourth year, 22.1% in the fifth, a slightly lower percentage (18.0%) in the sixth, 20.2% in the seventh, 15.5% in the eighth, 5.5% in the ninth, and 1.3% in the tenth. The relationship between the cumulative percentage of hybrids and the year of fruiting indicated the length of the juvenile stage in apple hybrids (seedlings on their own roots). It should be noted that only hybrids that began bearing fruit within the first 10 years of life were examined. However, some of the hybrids analyzed did not bear fruit at the end of the study period (probably due to an extremely long juvenile period or due to sterility). In the direct crosses, hybrids with a high number of fruits per tree were recorded in the following combinations: Reinette Baumann × Golden spur, Cluj Red × Kaltherer Böhmer, Roșu de Cluj × Ancuța, Aromat de vară × Melba, and Golden Delicious × Ancuța (Table 4).

These hybrid populations have the potential for successful selection of elite plants with profuse fruiting, but this trait must also be coupled to the size of the fruits and other advantageous qualities. The coefficient of variation per experiment (70.9%) demonstrated that the number of fruits per tree in F_1_ apple hybrids was an extremely variable trait. Within the families, the CV% was very high and had limits between 31.5% (Roșu de Cluj × Ancuța) and 92.6% (Kinrei × Jonathan).

Hybrids with large fruits were obtained in the following families: Aromat de vară × Mutsu, Roșu de Cluj × Kaltherer Böhmer, Ardelean × Prima, Aromat de vară × Reinette Baumann, Aromat de vară × Melba, Kinrei × Ardelean, and Golden Delicious × Ancuța. The coefficient of variation (CV%) fluctuated strongly within the hybrid families between 16.2 and 69.3%.

The number of fruits on the tree in the testcrosses (Table A2) varied widely, and the coefficient of variation was very high (81.3%). In the case of Mutsu and Prima as testers, the F value did not reveal significant differences. Therefore, it was concluded that the number of fruits per tree in the two half-sibling families was not due to true differences between the maternal parents but to errors. The Ancuta tester highlighted DSF 7/68, X–6–3, Roșu de Cluj, X–17–16, and Golden Delicious mothers, all of which had a good combining ability for the number of fruits per plant. They can provide biological material conducive to effective selection in the direction of choosing individuals with good prospects in the breeding process and at the same time can be used as suitable parents for high productivity in new hybridization programs. The variation limits of the average scores for the number of fruits per hybrid when Starkrimson was used as the paternal tester oscillated between 1.14 and 7.75, thereby indicating the significant differences between the maternal parents for their general combining ability. It resulted in a high CV% value (74.6%) for all half-sibling families with Starkrimson as the tester.

For the average number of days required for fruit ripening of F_1_ hybrids from families resulting from direct hybridizations, a relatively large range of variation was recorded (Table 5). The amplitude was between 126.7 days (Mutsu × Roşu de Cluj) and 186.0 days (Ancuța × Golden spur). The minimum value was recorded in offspring from two winter varieties since many hybrids had fruit ripening toward the end of August to the beginning of September. The high standard deviation may indicate that the early ripening of the fruits in this cross probably had a subjective causality determined by the triploidy of the maternal parent. The mean of all hybrid families produced by crossing winter-ripening parents (160.6 days) illustrated that within them, an effective selection could be found to obtain some winter varieties. The hybrid combinations that can provide a useful biological material for the selection of individuals with a very early ripening of the fruits proved to be Ardelean × Clar alb and Prima × Ardelean. The coefficient of variation for the number of days required for fruit ripening in the F_1_ apple hybrids had a relatively low value compared to other characteristics (i.e., the cross-sectional area of the trunk).

Fruit quality was highlighted in 21 hybrid combinations; some had a significant proportion of offspring with an appropriate taste for the selection. As a result, the hybrid combinations of Prima × Feleac, Ardelean × Feleac, and Prima × Ardelean stood out. Mutsu × Roșu de Cluj, Ardelean × Prima, Roșu de Cluj × Feleac, Golden Delicious × Ancuţa, Roșu de Cluj × Ancuţa, and Kinrei × Jonathan represented other hybrid families with good-tasting fruits that may be of interest for selection.

The coefficient of variation for fruit taste in the analyzed families was relatively high, thereby indicating a large diversity of fruit taste in each family. These values suggested the presence of a wide range of genotypes that ranged from low-quality- to high-quality-tasting fruit hybrids. Some hybrids produced fruits with taste comparable to the cultivars in the assortment.

In hybridizations with the five testers, the number of days required by F_1_ hybrids to reach physiological fruit maturity varied significantly (Table A3) both among families that had a common tester and within each hybrid family.

The testcross hybridization of 13 apple varieties and selections with the Feleac variety used as paternal tester highlighted the late maturing hybrids from the combination III-VI-5-26 × Feleac, which had an average maturation period of 201.4 days. A late offspring (174.0 mean days of ripening) was produced when a summer variety (Aromat de vară) and a winter variety (Mutsu as tester) were crossed. It was interesting that by crossing with a winter selection (X–6–64), offspring with an early fruit ripening (110.0 days) were obtained. The hybrids with the latest ripening when Ancuța was used as a tester resulted from the crossing with the DSF 3/80 selection (200.8 days), and a large part of the descendants completed the fruit ripening in storage. As a paternal tester, Prima, which is an autumn variety, induced a relatively late fruit ripening in the progeny (average of 154.7 days of experiment); even though the seven genotypes used as maternal parents contained one (X–13–10) with summer ripening and three (X–21–20, Ardelean and X–5–52) with autumn ripening. Using Starkrimson as a paternal tester (winter variety), an average of 174.3 days was achieved, which was higher than the averages produced by the other testers (Feleac, Mutsu, Ancuța, and Prima).

When Feleac was used as paternal tester, the differences between the mean scores for the fruit taste of the hybrids in the 13 families were not statistically significant (Table A3). Hybrids with pleasant-tasting fruits were obtained in crosses with the Prima, Ardelean, Roșu de Cluj, X–17–19, III–VI–5–26, DSF 3/86, DSF 7/68, and X–9–70 varieties. The CV% of the taste of the fruits in the ensemble families with Feleac as a tester was 33.1%, and within the hybrid families it was between 21.0–70.7%. Ancuta as a tester revealed a good general combining ability for progenies with good fruit taste for the following genotypes: X–9–69, DSF 7/68, DSF 3/41, X–6–3, DSF 5/22, X–9–70, Golden Delicious, Roșu de Cluj, and DSF 5/67. When using the Prima variety as a tester, the possibility of identifying hybrids with very-good-tasting fruits was high in combinations with X–13–10 and X–5–65. Instead, as for fruit size, the worst results for fruit taste were obtained when crossing with the X–21–20 selection. As a tester, Starkrimson identified DSF 3/58, 218/2, DSF 5/45, and X–6–3 as good parents for fruit quality. In Starkrimson half-siblings, it was observed that the higher the average score in a hybrid family (indicating a high proportion of individuals with good-tasting fruit), the lower the variation in the trait, thereby indicating the relative uniformity of individuals with the trait.

In the ensemble of analyzed hybrids with edifying results for fruit development and ripening both from direct hybridizations and testcrosses (5484 hybrids), the following proportions were obtained for the ripening period of the fruits: summer 12.7%, autumn 53.5%, and winter 33.8% (Table 6).

The important genetic additive effects as well as the non-additive genetic effects in the transmission, fixation, and phenotypic manifestation of fruit ripening in F_1_ apple hybrids were highlighted by combining all hybridizations in a diallel system with three groups of parents according to the season of fruit ripening (summer, autumn, and winter) (Table 7). The results showed that if parents with a certain fruit maturity were used, the additivity effects contributed significantly to obtain offspring with the same maturity. The general combining ability (GCA) value for the summer cultivars was significantly positive (+1.7457 ***), which indicated that they were the most suitable parents for obtaining offspring in which those with a summer ripening of fruits predominated. The same situation was recorded in the autumn and winter parents. Instead, due to significantly negative GCA values, the additivity of polygenes acted against producing hybrids with summer fruit ripening when the parents were represented by autumn and winter varieties.

Interaction effects of dominance and epistasis were significantly positive in crosses with parental forms of the summer × summer type (CSC = +1.1043 **). Contrary, in summer × autumn and summer × winter hybridizations, the non-additive effects significantly prevented the obtaining of offspring with a summer fruit ripening. Thus, significant additive and non-additive effects were involved in the genetic inheritance of the summer ripening of fruits (the CSC constancy had high positive values in maternal parents with summer ripening of fruits). Therefore, summer fruit ripening was determined either by the combined effect of additivity and dominance and epistasis interactions or solely by the no-additive interactions. These latter effects, although partially non-transmissible, could decisively influence the trait in F_1_ hybrids, especially in summer × summer crosses.

Apple scab (*Venturia inaequalis* (Cke.) Wint.) attack on F_1_ apple hybrids from 22 direct hybrid combinations highlighted relatively low values of the average scores recorded for most hybrid combinations (Table 8). However, significant differences between the tested families were identified. The most sensitive hybrids to the apple scab attack were those from the combinations Mutsu × Roşu de Cluj and Ancuţa × Starkrimson. Instead, the hybrids from the crosses Ancuţa × Mutsu, Ancuţa × Roşu de Cluj, and Prima × Feleac showed a proper response to the apple scab attack. Descendants with a certain tolerance to the disease were also obtained in the following combinations: Aromat de vară × Reinette Baumann, Prima × Ardelean, Ardelean × Feleac, Roşu de Cluj × Feleac, Aromat de vară × Mutsu, and Reinette Baumann × Golden spur. Obviously, the chances of selection for the identification of promising elites were increased in hybrid populations that stood out due to resistant or tolerant trees to apple scab attack.

The hybrids of Aromat de vară × Mutsu, Aromat de vară × Ancuta, and Ancuta × Starkrimson provided an appropriate response to the powdery mildew (*Podosphaera leucotricha* (Ell. et Ev.) Salm.) attack. Within the hybrid families, the coefficient of variation of the notes for disease response ranged from small-medium (CV% of 10.2% in Roşu de Cluj × Ancuța) to very high (CV% of 52.8% in Ardelean × Prima). The CV% of powdery mildew noted in the experiment represented by the 22 hybrid combinations was 38.1%.

The average notes between half-sibling families based on the tester did not differ significantly for the apple scab attack in the testcrosses, but there were significant differences in each of the five types of testcrosses (Table A4). Each tester highlighted genotypes with a higher general combining ability to produce F_1_ hybrids with a suitable response to apple scab. Thus, in the hybridizations with Feleac, the following stood out: X–17–19, Prima, X–9–19, and Roşu de Cluj (1.35). With Mutsu as a tester, less sensitive hybrids to scab fungus were recorded when the mothers were Ancuța, Aromat de vară, and X–5–71, while the Ancuța tester highlighted the X–17–16 and X–21–20 selections. Prima as the tester highlighted X–5–65 and X–21–20 selections, and Starkrimson two other selections: III–II–17–25 and DSF 1/54.

The testcross schemes were also useful in identifying hybrid families in which the selection for resistance or tolerance to powdery mildew could be more efficient. Feleac as a tester indicated the hybrid populations in which the mothers were X–5–71, X–3–8, X–9–70, DSF 7/68, and III–VI–5–26. Mutsu highlighted Aromat de vară, 218/2, X–6–73, and DSF 3/70; Ancuța as the tester highlighted DSF 5/67, X–17–16, and X–9–70. Prima identified the following genotypes: X–5–65, X–21–20, X–6–24, and X–5–52; while Starkrimson as the tester highlighted 218/2, DSF 3/58, and X–6–3.

The main genetic parameters analyzed within half-sibling families depending on testcrosses are presented in Table 9. There were clear differences between the mean values obtained in the F_1_ hybrids in the half-sibling families depending on the trait and tester used. The genotypic coefficient of variation (GCV) registered large oscillations depending on the tester for the following traits: fruit size—between 2.6 (Feleac) and 41.8 (Prima); fruit taste—between 8.1 (Mutsu) and 49.8 (Prima); and response to powdery mildew attack—between 9.8 (Mutsu) and 41.0 (Prima). For the response of the F_1_ hybrids to apple scab attack, because the Prima cultivar has a monogenic resistance to disease (*Vf* gene; i.e., *Rvi6*), the GCV value is only indicative and presented in parentheses. For the same reason, heritability coefficients were also not calculated.

There were differences between heritability in the broad sense (H^2^) and heritability in the narrow sense (h^2^) depending on the trait and the tester. For H^2^, a greater variation in the genotype’s contribution to the trait’s phenotypic expression was observed as follows: fruit size—between 0.531–0.946 (Feleac and Prima); taste of the fruits—between 0.566–0.975, and response to powdery mildew attack—between 0.554–0.891 (Mutsu and Prima). Greater uniformity of H^2^ values among testers was identified for fruit ripening and tree vigor (TCSA and tree height).

The genetic effects of additivity played a different role depending on both the tester and the trait analyzed. Thus, the non-additive genetic effects of dominance and epistasis had a more consistent role for TCSA with Starkrimson (h^2^ = 0.408), for tree height with Mutsu (0.416), for fruit size and fruit taste with Prima (0.384 and 0.410, respectively), for fruit ripening with Starkrimson (0.427), and for apple scab and powdery mildew attack with Starkrimson (0.341 and 0.376, respectively). Expected response to selection had relatively high values with Starkrimson for TCSA and the number of fruits per tree and with Prima for fruit size, fruit taste, and response to powdery mildew attack.

The percentage of F_1_ apple hybrids that presented the analyzed traits expressed at an optimal level for selection differed strongly depending on the character of interest for apple breeding as well as on the type of hybridization in which the estimation was performed (Table 10).

A relatively low percentage (4.6%) was obtained for fruit flavor in direct hybridizations, but an unexpectedly high percentage for the trees’ tolerance to apple scab attack was obtained. In the case of the testers, the percentage of F_1_ hybrids with very good fruit taste was also very low, especially with Starkrimson (1.4%) but also with Ancuța and Prima and slightly higher with Feleac (9.2%). Overall, a higher proportion of hybrids appropriate for selection were obtained under the hybridization categories for winter ripe fruits, apple scab tolerance, and powdery mildew. Among the testers noted were Ancuța and Feleac (for the ‘large fruit’ trait), Starkrimson and Prima (for tolerance to apple scab), and Mutsu (for weak vigor of the trees).

A principal component analysis (PCA), a multivariate technique for analyzing quantitative data, highlighted a close link between hybridizations when utilizing Mutsu as the tester and direct hybridizations (Figure 3). Ancuța was also positioned in the same quadrant (quadrant I, top right) but at a certain distance. In addition, the other testers were on the right side of the vertical axis of the PCA but in quadrant II (lower right). The Prima cultivar was found in the area that was the farthest from the other places (particularly in relation to Mutsu).

Tolerance to apple scab was the only trait that was placed in a different quadrant from the others. This trait was the most different from the others (quadrant I). In opposition to the adequate response of trees to apple scab attack, the main characteristics of fruit production and fruit quality appeared, namely the number of fruits per tree, large fruits, and high eating taste. On the other diagonal, in quadrant IV there was reduced tree vigor (trunk section area and tree height, both elements of tree growth that were located close) in an inverse relationship with winter ripeness of fruits and tree tolerance to powdery mildew attack. Of the two main components of the PCA, PC1 captured a very large part of the total variation (90.5%) but PC2 only a small part (6.2%).

A UPGMA dendrogram (Figure 4) confirmed the distance of tolerance to apple scab attack from the rest of the analyzed characters, which was highlighted previously by PCA. The tolerance to apple scab formed as a distinct cluster a singular character that was different from the other cluster, on which there were two subclusters: one with three characteristics and the others with four characteristics grouped by two.

If the arrangement of the vigor elements of the trees together in a common subcluster was to be expected, the positioning of a high number of fruits per tree and high eating taste in a common subcluster appeared quite surprising. The close relationship revealed by the multivariate analysis between these two traits may suggest that hybrids with a larger fruit load in the crown of the trees may produce fruits with a better taste. Similarly, another cluster with two close characters could indicate that the winter ripening of fruits can be associated with larger fruits. These two characteristics would be quite closely related to tolerance to powdery mildew attack.

Extreme variation existed in the minimum of progeny required to produce a hybrid with quantitative traits of agronomic interest (Table 11). Regardless, this computed amount (if these traits behave independently) appeared extremely high depending on the type of hybridization (direct vs. testcross), the tester, and the cumulative traits of interest.

The average for the entire experiment was at least 122 offspring for the two traits that were crucial in the selection of elite plants (high eating taste and large fruits) and increased to 692 progenies if two more traits of great interest in apple breeding were added (tolerance to apple scab and powdery mildew attacks). If we added a fifth trait (high number of fruits per tree), we reached 6383 hybrids, and then the exponential growth continued to impressive numbers with the addition of the next desired traits.

Thus, if we combined the intended fruit ripening period (a breeding goal in HRS was late ripening to produce ‘winter’ cultivars) with the lowered tree vigor (another goal was to produce cultivars suitable for crop intensification), we obtained 259,509 hybrids. Interestingly, if instead of weak vigor as trunk cross-sectional area we included weak vigor as tree height, the minimum number of hybrids was reduced to 166,835. Analysis indicated that among the testers, Feleac and Prima produced the best results for the first five extremely relevant traits (high eating taste, large fruits, tolerances to apple scab and powdery mildew attack, and high number of fruits per tree).

## 3. Discussion

### 3.1. Considerations Regarding the Results Obtained for Quantitative Traits of Interest in Apple Breeding

The analysis of F_1_ hybrid populations derived from direct and testcross hybridizations highlighted the existing differences in the important quantitative traits of seedlings both between families and within the families and depending on the parents and their crossing. The inheritance of the investigated traits followed a predictable quantitative pattern, and the offspring genetically inherited the traits of the parents.

Our results confirmed that the polygenic inheritance of tree vigor [43,44,45,46] does not exclude possible maternal (cytoplasmic) effects, which is also the case for inheritance of precocity [47]. Possible influences of extranuclear inheritance appeared noticeable in our direct and reciprocal crosses; i.e., Prima × Ardelean and Ardelean × Prima. The resulting variation in most hybrid combinations allowed successful selection to identify elite plants with low vigor. As reduced tree vigor remains a current breeding objective [48], selection in families in which progeny have lower vigor is recommended to ensure high efficiency. In some combinations, there were a few discrepancies between the vigor of the parents and the vigor of the F_1_ hybrids (e.g., in testcross with Mutsu, which probably was due to triploidy of this cultivar, and with Starkrimson due to the specific combining ability and non-allelic interactions of dominance and of epistasis).

Even if tree vigor is considered a quantitative trait [43,44], the type of growth and branching or habitus of the trees is sometimes a monogenic trait [49,50]. However, tree growth has a complex character because the architectural ideotype of the trees and its vigor influence each other. The rootstock, tree management system, and cutting influence the phenotype of the trees in the orchard [51]. As a result, an accurate assessment of the natural type of growth and branching can be made in the hybrids on their own roots that have not been trained or pruned. The distribution of our F_1_ descendants from the experiment in a certain type of growth and fruiting demonstrated the reduced proportion of spur, weeping, and columnar architectural ideotypes compared to ‘standard’. The spur type included a reduced vigor of the trees and particularities useful for intensive culture that are much desired in modern apple breeding programs [17,51]. The columnar (compact) habitus is inherited by a dominant gene (*Co*) [50,52], but in our hybridizations such parents did not participate. The proportion of columnar seedlings from the total of hybrids was close to the value obtained in a similar study containing hybrids of various origins [53]. If the columnar ideotype was determined by the *Co* dominant gene [54], in hybrid combinations with columnar progeny at least one of the parents should have been heterozygous for the *Co* allele. This is unlikely because it would have resulted in Mendelian segregation of the columnar offspring. Therefore, there was a greater probability for the hypothesis of a polygenic inheritance of the columnar ideotype because even ‘common’ spur-type varieties can produce offspring with a compact growth that is almost as extreme as the Wijcik type [55] but with a moderate or low frequency [56]. The results did not exclude the hypothesis that columnar growth appears as a double recessive trait associated with a strong reduction in vigor [53] or implication of genes with pleiotropic effects [57]. Spur, compact, and columnar types can sometimes be identified in a single hybrid progeny of a single cross [53]. Similar situations could arise for weeping (pendulous) phenotypes even if it is known that this architectural habitus is controlled by a single dominant allele (*W*) [49,58].

The juvenile period is of great interest in apple breeding because juvenility delays the selection process, increases the costs of tree maintenance, and occupies the land for a certain period [51,59,60]. In the inheritance of the juvenile period, apart from the transmissible additive effects of polygenes (which are predominant), genetic effects of dominance and epistasis are also involved as well as maternal effects [61,62]. The hybrids that did not fruit early in the field (e.g., until the sixth year) were considered to be economically unviable and were indicated to be eliminated [63]. Such individuals are unlikely to be valuable for breeding especially when considering the direct correlation between the length of the juvenile period and the delaying of fruiting after grafting [17,64].

The size of the fruits along with the number of fruits per tree provided valuable information about the production potential of offspring from various types of crosses subjected to the selection process [56]. Some of the hybridizations produced offspring with appropriate fruit size as well as good fruiting potential. It was confirmed that the polygenic nature of fruit size [45,65] was strongly influenced by environmental and cultural conditions but to a different extent that depended on the genotype of the parents and the crosses performed. In half-sib families of Prima as the tester, the influence of environmental factors on fruit size was significantly lower compared to that in half-sibling families of Feleac, Ancuța, Mutsu, and Starkrimson. Although all parents had large fruits, the average fruit size was lower in the offspring. The phenomenon is explainable when considering the selection pressure exerted over time for this trait [51] and even if it is assumed that the domestication of the apple began with a great advantage and a much lower evolutionary pressure than other cultures [66,67]. Fruit size in elites selected from hybrids with mid-sized fruits can be improved with appropriate rootstock and superior agrotechnics [17].

Breeding objectives aimed at enriching the assortment of apples with varieties of different ripening to ensure fresh fruit throughout the year have lost interest. If in the past the goal was to develop early summer to winter types with good fruit preservation in storage, the significance of these objectives has been diminished by modern storage conditions [51]. Our results were consistent with the polygenic inheritance of fruit ripening [68,69], thereby confirming that due to the additivity of polygenes by using suitable parents, F_1_ hybrid populations could be obtained in which hybrids with the desired ripening period predominated and the selection in the respective direction was efficient [70]. This applied even if in winter × winter crosses the proportion of offspring with late fruit ripening was relatively low (43.5%). However, when compared to other traits (i.e., trunk cross-sectional area and number of fruits per tree), fruit ripening in F_1_ apple hybrids appeared as a less variable character and as a result were relatively easier to transmit and fixed in the offspring. Apple fruit ripening as well as the flowering period are quantitatively inherited, and deciphering their genetic control is essential for breeding cultivars adapted to growing environments [71].

The taste of the fruits remains a crucial characteristic in the evaluation of hybrid populations and the selection of elites [17,51]. Although all parents in our hybridizations possessed adequate fruit quality, genetic recombination in the progeny was manifested by a wide variation in fruit taste (from poor to good). The hybridizations provided a useful biological material for the selection of elites with high fruit quality. Some parents stood out for their high number of hybrids with quality fruits (e.g., Ardelean, Prima, Feleac, Roşu de Cluj, and Ancuţa). It must be stated that not every hybrid combination of these varieties provided valuable descendants for the quality of the fruits. As a result, it is recommended to use in crosses some parents that not only have fruits of good organoleptic quality but also the combinative ability necessary for the transmission of this trait to hybrids. However, due to this well-known fact, it reached the excessive use of a limited number of cultivars (‘professional parents’ crossed as ‘good’ × ‘good’), which led to the risk of narrowing the genetic base of apple cultivars [15].

Among the apple diseases, *Venturia inaequalis* (Cke.) Wint., which causes apple scab, and *Podosphaera leucotricha* (Ell. et Ev.) Salm., which causes powdery mildew, are the pathogens that causes the greatest damage to apple crops [72,73]. Attack variation among F_1_ offspring was generally high, thereby indicating the presence in most of the analyzed combinations of some plants with a different response to the pathogens: from not attacked or weakly attacked (resistant or tolerant) to medium or strongly attacked (susceptible or even very sensitive to the disease). Testcrosses highlighted cultivars and selections with good combinatory ability to obtain resistant or tolerant progeny to both diseases.

It is well known that Prima was the first apple cultivar in a series to carry the *Vf* (*Rvi6*) gene for resistance to apple scab derived from *Malus floribunda* 821 [74]. Due to the dominant gene *Vf,* which was transmitted to approximately half of its offspring, Prima produced hybrids with an average score for apple scab attack that was halved compared to the other testers. Because of monogenic resistance, the genetic parameters specific to the characters with polygenic inheritance were not calculated for Prima half-siblings. It was obvious that obtaining hybrids with good resistance or at least tolerance to apple scab attack was relatively easy to achieve by using suitable parents (or so-called ‘resistant’ parents) [75,76,77]. Crossing Prima with selections that possessed the same *Vf* gene (e.g., from the New Jersey (N.J.) group) resulted in obtaining a higher percentage of hybrids with genetic resistance to apple scab according to Mendelian reports. Hybrids with a proper response to apple scab attack were obtained by crossing the X–21–20 selection, which was derived from a cross between an interspecific hybrid of Reinette Baumann × *Malus niedzwetzkyana* (see Appendix A) with Prima. Interspecific hybrids are likely to inherit vertical resistance from Prima (or other sources with monogenic resistance) and horizontal resistance from wild species [78]. It is known that obtaining new apple cultivars that incorporate vertical and horizontal genetic resistance would confer increased resistance to the attack of the pathogen. Identifying and mapping resistance genes will assist in creating new apple cultivars with genetic pyramids for various and durable resistances [79]. When considering the response to apple scab as a quantitative character, the additivity had a more substantial contribution when the testers were represented by the Ancuța and Starkrimson cultivars compared to Feleac and Mutsu.

In the direct hybridizations, offspring with reduced susceptibility to powdery mildew were obtained in the following crosses: Aromat de vară × Mutsu, Aromat de vară × Ancuţa, and Ancuţa × Starkrimson. Some parents (Aromat de vară and Ancuța) produced hybrids that were both resistant (or tolerant) and sensitive to powdery mildew, which illustrated the important role of the hybridization partner or the hybrid combination. The difference between direct and reciprocal hybrids of Ancuța and Roşu de Cluj led to the hypothesis that the sensitivity to powdery mildew of Roşu de Cluj (previously reported as a carrier of polygenic resistance to apple scab [80]) is transmitted more faithfully as the maternal parent. Among the testers, Prima produced the offspring with the best response to powdery mildew and was appreciated as a good parent not only for resistance to apple scab but also for resistance or tolerance to powdery mildew. These results confirmed that resistance to apple scab and powdery mildew can be further improved by the judicious selection of parents involved in future hybridizations [78,81,82].

### 3.2. Considerations Regarding the Complexity of Apple Breeding and the Theoretical Possibilities of Applying Selection in Hybrid Populations

The broad- and narrow-sense heritability calculated in this study varied within the limits of the majority of apple breeding experiments for normally distributed variables. Like in other investigations, the parental forms, the type of cross-breeding, the traits examined, and their scale influenced the heritability values (i.e., absolute values for TCSA and tree height, mark scales with different gradings for other traits, etc.). Overall, it was confirmed that the additivity effects of polygenes were more important than the non-additive effects [44,46,61]. However, dominance and epistasis can contribute to the overall genetic variance of traits of interest in apples in a consistent proportion. Among the quantitative traits studied in current research, the most dramatic effect in reducing the number of individuals suitable for selection was the good taste of the fruits. Unlike other characteristics, fruit taste is an essential element for choosing a hybrid as an elite plant. Fruit size is also essential in apple selection, and this trait was expressed at a corresponding level in our F_1_ hybrids (in about one-fourth of the offspring). It was found that the statistical size of the hybrid population required to find at least one hybrid that possessed all of the desirable characteristics at a high level increased exponentially with the number of traits of interest associated with that hybrid (and subsequently a new variety). As an average for all hybrids, this number was 122 for two traits (high eating taste and large fruits), 692 for four traits (if resistance/tolerance to apple scab and to powdery mildew attack were added to the first two), and 6383 (if the fifth trait (the high number of fruits per tree) was added). If the winter ripening of fruits was also added, a very high value of 21,292 hybrids needed was reached. Adding low tree vigor resulted in 259,697 hybrids (when considering TCSA) and 166,872 hybrids (when considering tree height). A similar study has not been reported since the one conducted by Williams, who showed that one of 6250 hybrids will possess a combination of five quantitative traits independently from one another with a reasonable level of expression [83].

The results regarding the complexity of the breeding process through hybridization and the extremely large number of descendants required for the selection of valuable descendants, which accumulate favorable characteristics expressed at a higher level [84,85], appeared downright discouraging for the breeder. Out of the total number of hybrids tested in the field based on the mentioned selection criteria, the hybrids chosen as elite represented 3.52%. Further, half of this percentage was grafted, and the resulting clonal selections were subjected to testing in the control fields. However, the data presented referred to the entire set of hybridizations performed whether they were direct or testcrosses, into which all the families obtained entered without regard to the utility or value for the selection of individuals. If the families that produced offspring worthless for selection had been eliminated and only combinations had remained that contained individuals with as many favorable characteristics as possible, surely the results would have been much improved. In addition, the genetic recombination of the F_1_ phenotypes probably was influenced due to a certain inbreeding of the parents involved obtained at HRS. As shown in Appendix A, Ardelean, Ancuța, Aromat de vară, Feleac, and Roşu de Cluj had a common parent (Jonathan). Utilizing related parents in hybridizations therefore decreases genetic diversity, increases allele loss through genetic drift, and decreases within-family variance and potential genetic gain [86]. Selecting families based on the GCA values of the parents can substantially reduce the size of hybrid populations and increase the efficiency of selection [87]. In apple breeding, recurrent selection for GCA to genetically improve hybrid populations is also effective [87,88,89,90,91]. However, hybrid combinations with a low possibility of selecting promising elites are small and are not economically sustainable. Therefore, such combinations should be avoided by carefully selecting the parents involved. If the breeding objectives include resistance to diseases, the efficiency of the breeding increases by using appropriate parents (resistant to apple scab, powdery mildew, etc.) [92,93]. In addition, the selection is simple to apply in the phase of 3–4 leaves via artificial infections conducted in the greenhouse. Thus, costs can be significantly reduced because only the appropriate hybrids will be examined further in the field until the elite plants are selected [82]. The selection’s efficiency is directly related to the appropriate choice of parents as well as the selection criteria and the number of characters followed [51,78,84,94].

### 3.3. Considerations Regarding Apple Breeding Research and the Significance of Introducing New Cultivars

If 10,000 varieties of wheat were cultivated in China in 1949 and by the 1970s the number was reduced by 90%, Goland and Bauer [95] showed that the same situation was repeated with other major crops as well as vegetables and fruits, including apple. Numerous factors contribute to the loss of crop biodiversity and increased genetic erosion, including agricultural industrialization, Green Revolution technologies, environmental changes, civil conflicts, changing market characteristics (including distance to market), the domination of crops by few varieties, etc. [96,97].

Internationally, the multitude of cultivars indicates how significant apple culture is in terms of food, nutrition, diet, health, social, and economic considerations [1,3,18,34]. However, although there are over 10,000 varieties of apple in the world, relatively few are widely grown and cultivated [98,99]. There are opinions that there are even more than 30,000 varieties of apples in the world [100], of which only 20–40 are cultivated and traded commercially. In 2007, only 13 varieties were grown on approximately 72% of the total apple area in Europe. The numerous varieties of pome trees can be artificially influenced by old or heirloom accessions and locally grown varieties with various names, homonyms, and synonyms [35,101]. Moreover, a large part of the world’s apple production is obtained from a small number of varieties because only a few of the large number of cultivars are grown over large areas [17,102]. While the genus *Malus* has a wide genetic diversity, cultivated apples have a narrow genetic background due to the common origin of many cultivars, the selection process, and heterozygosity maintained by continuous vegetative propagation [45,103]. Many apple cultivars with a significant extension in culture have a common genetic basis because they were obtained by selecting natural mutations such as bud sports or from chance seedlings obtained from the best varieties and widespread in the world; in fact, the apple, like other cultivated species of great economic importance, has a narrow genetic background, and due to this it can be vulnerable to a catastrophe at any time [17,36,98,104].

The widespread use of the resistance gene *Rvi6* (*Vf*) to scab (*Venturia inaequalis*) from ornamental or Japanese apple (*Malus floribunda*) [105,106,107,108] accelerated the narrowing of the genetic base of new selections and varieties that incorporate this gene [109]. More than 80% of the scab-resistant cultivars released carry the *Vf* gene [110]. Many apple breeders believe that limiting the genetic resources used in apple breeding lately can have unfavorable consequences on the cultivated species [102]. To improve the genetic background in apple breeding, different methods can be used; i.e., interspecific hybridizations or recurrent selection for general combining ability (GCA), but these require many resources, high costs, and a long time for recovery [87,88,111].

Climate change, the intensification of agriculture, the increase in the number of phytosanitary treatments and the diversification of products to combat diseases and pests, and the emergence of new pathogens or insects or their physiological or virulent races and strains accentuate the vulnerability of the apple to new challenges [75]. Proposed ways to avoid crisis situations in apple culture include the acquisition and conservation of genetic resources in the germplasm pool, the assessment of existing variation, and apple breeding and anticipatory strategies. Utilizing the genetic potential of the *Malus* genus as effectively as possible and obtaining ‘durable’ varieties with a good response against diseases and pests, biological potential, and ecological adaptability combined with the very high quality of fruits can result in the creation of new varieties [3,112,113]. In Romania, apple culture has an old tradition, and remarkable results have been achieved in apple breeding [114,115]. In addition, concerns have also focused on the amplification of the gene pool required for selection by both intraspecific and interspecific hybridization [45,111]. HRS served as a model organization for horticultural research and contributed significantly to the development of new cultivars, the preservation and assessment of genetic resources, and the production of planting material. The unit contributed to the advancement of fruit growing by supporting farmers as well as the education and training of personnel. The funding of research became unstable when the communist system fell in 1989, and just 5 of the 26 fruit research stations in the Research Institute for Fruit Growing Pitesti (RIFGP) network were still functioning. There was a loss of experimental fields, biological materials, genetic resources, and human resources. There is hope for HRS with the transition to the agricultural university of Cluj-Napoca if national projects allocated to agricultural research are initiated. Like everywhere else, scientists are hoping that those in authority will reconsider the importance of plant breeding and the cultivar’s significance as the most crucial production and crop component.

Finally, based on the findings of this study, it can be concluded that the hybrids from the combinations with the highest scores for fruit taste provided the greatest proportion of clonal selections from the investigated hybrid populations. Some of these selections were obtained not just from ancestors represented by high-quality varieties but also from offspring obtained through free pollination. Because the open pollination in the latest case occurred in experimental fields with thousands of genotypes (species, cultivars, and hybrids), it is probable that in the future they could represent a broad genetic base beneficial to new breeding projects. In addition, favorable outcomes may be produced in the future by combining the valuable traits from parents represented by these clonal selections and creating double or complex hybrids, thereby diversifying the genetic background of the new cultivars.

## 4. Materials and Methods

### 4.1. Location and Conditions of the Study

The research was carried out at the Horticultural Research Station (HRS) in Cluj-Napoca, North West Romania. The station was founded in 1953 and has produced more than 20 new apple and pear cultivars. The main apple breeding objectives were reduced tree vigor, productivity, fruit quality, and disease resistance. The HRS pome tree breeding sector produced 10,000–20,000 hybrids annually and followed selection procedures in the field (Figure 5). The experimental fields of HRS are located on a degraded chernozem-type soil with a medium supply of the main chemical elements. The land has a slight slope with a southern exposure and is suitable for both the growth and fruiting of plants. The favorability of the soil and the general conditions in the field experiment were specific to a characteristic area of the Someș Mic Valley Corridor, Cluj County [116]. The average annual temperature is 8.2 °C and the average annual precipitation is 560 mm.

Site and soil preparation, fertilization, and planting were standard and similar to those recommended in commercial orchards. However, due to a lack of funding and resources allocated to HRS in the last two decades, phytosanitary treatments, fertilization, and pruning or other interventions were not applied after planting to the hybrids included in this study. Thus, the hybrids had a natural vegetative growth, branching, crown, and fruiting; consequently, tree architecture, fruiting, and disease attacks were evaluated in the field without technological and phytosanitary interventions.

### 4.2. Biological Material

All F_1_ apple hybrids in this study were obtained from intraspecific crosses with different cultivars or selections as parents as part of a breeding cycle in a standard and traditional scheme described by Wannemuehler et al. [117]. The data recorded for more than 5000 apple hybrids on their own roots analyzed in the fields in the selection process and the choosing of the elites were processed. The hybrids included in the study were obtained from direct hybridizations and testcrosses depending on how the parents participated in the hybridizations (Figure 6) and benefited from the same conditions.

In the direct hybridizations, hybrids were originated from 21–25 hybrid combinations (families). The parents were some well-known apple cultivars that included Jonathan, Golden Delicious, Starkrimson, Starking, Reinette Baumann, Kaltherer Böhmer, Melba, Golden spur, Mutsu, Prima, etc.; or cultivars created at HRS: Roşu de Cluj (in English: red of Cluj), Aromat de vară (in English: flavored of summer), Feleac, Ancuţa, and Ardelean. Some hybrids resulted from direct and reciprocal crosses (e.g., Prima × Ardelean, Ardelean × Prima, Aromat de vară × Mutsu, and Mutsu × Aromat de vară). F_1_ hybrids from testcrosses were derived from crossing a variety used as a paternal tester with different cultivars or selections (elites) obtained at HRS used as maternal parents. In the five testcrosses, the testers used were represented by the following cultivars: Feleac, Mutsu (included as an exception in the experiment for data comparison because it is a triploid variety), Ancuţa, Prima, and Starkrimson. Appendix A shows the origin of these cultivars obtained at HRS, while Appendix A provide the origin of the selections involved in hybridizations.

### 4.3. Procedures of Performing Observations and Measurements

In the study, the data recorded in HRS observation registers were processed. In each hybrid family, the number of offspring as individuals (plants) was extremely variable and ranged from 7 to 178 seedlings. There were no seedlings removed before the data were recorded. All the offspring of the examined hybrid combinations were considered for the study regardless of their positive or negative traits. Due to the limited planting distances (3.0 m between rows and 1.0 m between plants per row) in the hybrid fields due to the terrain economy, the hybrids had an obvious tendency to increase in height. Although the branching of the trees was also influenced, it was assumed that all hybrids had the same uniformity of conditions. Due to the absence of tree cutting and fungicide treatments, the evaluation of the response of the hybrids to the attack of diseases was performed under natural conditions of infection using data from all years processed. The evaluation of the hybrids was performed based on the UPOV recommendations [41] and the methodology used in fruit tree breeding in Romania [63].

### 4.4. Observations, Measurements, and Determinations

The observations, measurements, and inferences were focused on the key elements of growth vigor, yield, fruit quality, and response to the main diseases of the species. In general, these factors are also considered while choosing elite plants for hybrid fields. Furthermore, assessments were made on the length of the juvenile stage, the type of growth and fruiting, and the fruit’s ripening time.

In order to assess the vigor of plant growth, measurements were made of the height and thickness of the trunk. The study considered data registered at the end of the 12th year of life in the autumn after the hybrids had finished their vegetation. The height of the trees was measured from the ground level to the top of the trees and expressed in meters; the diameter of the trunk was measured at 20 cm above ground in the row direction. Based on these data, the trunk cross-sectional area (TCSA) was calculated and expressed in cm^2^. Depending on the height and the TCSA, the vigor of the hybrids was arbitrarily considered as low (less than 3.5 m or 38.5 cm^2^), medium (between 3.5–4.5 m or 38.5–78.5 cm^2^), or strong (over 4.5 m or 78.5 m^2^).

After the hybrids entered the fruitification period, their productivity was analyzed via observations made on two essential elements of productivity: the number of fruits per plant and the size of the fruits. The number of fruits per tree was assessed by grading them with notes as follows: 1 = small number of fruits per tree (less than 6–8 regardless of fruit size); 4 = average number of fruits per tree (between 6–8 and 20–30); 7 = large number of fruits per tree (between 20–30 and 60–70); and 10 = very large number of fruits per tree (over 60–70). The size of the fruits was also assessed by grading them (for the same reasons of expeditiousness) with notes as follows: 1 = very small fruits (less than 50 g); 4 = small fruits (between 50–85 g); 7 = medium-sized fruits (between 85–125 g); and 10 = large fruits (over 125 g). For the ripening period of the fruit, the number of days from the beginning of flowering to the physiological maturity of the fruit was considered. The ripening of the fruit was considered to be summer if the apples had reached the maturity of consumption until September 1; autumn if the optimal time of consumption and fruits preserved capacity period was in the autumn months of September, October, and November; and winter if the optimal time of consumption and fruits preserved capacity period was in the winter months (even in the spring). In the assessment of ripening for late autumn and especially winter apples, the storage time (cellar without special conditions) was also considered and was given as the number of days from the beginning of flowering to the moment when the fruits reached optimal consumption maturity.

The taste of the fruit was appreciated by tasting and grading with the following notes: 2 = weak; 6 = medium; and 10 = good. The hybrids marked with 10 bore fruit at the quality level of most cultivated varieties, including very good or excellent quality. Those marked with 6 had fruits below the level of the lowest rated varieties in terms of the taste quality of apples, and those in grade 2 had fruits that were unfit for consumption (non-edible).

The response of hybrids to the attack of the main diseases of the species—apple scab (*Venturia inaequalis*) and powdery mildew (*Podosphaera leucotricha*)—was appreciated in the natural conditions of infection; in the hybrid fields, the fungicide treatments was excluded. Each year, scab and powdery mildew incidence was assessed two times: in the first week of July and August and based on scab attack on leaves and powdery mildew on shoots. For expedition, a scale of 0 to 5 was used that followed the standard diagram corresponding to an attack index or attack degree (AD%) as follows: 0 = no attack, which represented a ‘zero’ apple scab and/or powdery mildew attack degree (AD% = 0); 1 = very weak attack (AD% = 0.1–1); 2 = weak attack (AD% = 1.1–5.0); 3 = medium attack (AD% = 5.1–15); 4 = strong attack (AD% = 15.1–20); and 5 = very powerful attack (AD% > 20.1) [111]. The data were processed as average values per individual and per year and then on hybrid combinations and the entire experiment.

### 4.5. Statistical Analysis

The data were summarized as means and standard deviations for each analyzed trait and hybrid combination. Statistical analysis of the experimental data was performed by applying the analysis of variance (ANOVA). When the value of the ratio F was greater than the appropriate critical distribution F at α = 0.05, the null hypothesis was rejected when considering that at least one of the means was significantly different from the other means. In this case, a post hoc test using Duncan’s multiple range test (α = 0.05) was performed.

Both broad-sense heritability (H^2^) and narrow-sense heritability (h^2^) were calculated, and genetic variances were partitioned based on the relationship between the half-siblings [118,119,120]. Additive, dominant, and epistatic variances were used to divide the entire genetic variance into its three components (V_G_ = V_GA_ + V_GD_ + V_GI_). The genetic variance, environmental variance, and genetic × environmental interaction variances were divided into the overall phenotypic variance’s constituent parts (V_P_ = V_GA_ + V_GD_ + V_GI_ + V_E_ + V_G*E_). All siblings were thought to have exposure to the same technology and life environment. Using the equation H^2^ = V_G_/V_P_ (where V_G_ represents genotypic variation and V_P_ represents phenotypic variance), broad-sense heritability was computed. According to the equation h^2^ = V_GA_/V_P_ (where V_GA_ stands for additive genetic variance and V_P_ stands for phenotypic variance), narrow-sense heritability was computed. Methods based on half-sibling variance were used to calculate H^2^ and h^2^; this took into consideration that the proportion of half-siblings’ shared genes (or their degree of relatedness) was equal to 25% (or 1/4 of the additive variance) [118].

The genotypic coefficient of variation (GCV) was calculated by dividing the genotypic standard deviation (S_G_) by the trait mean of half-sib families depending on the tester (GCV = S_G_/$\bar{x}$ × 100), and heritability in a narrow sense was used to predict the response to selection (R) using the formula $R=i\times \sigma_{P}^{2}\times h^{2\;}$ (where *i* = selection intensity (the value was 2.06 at a 5% selection intensity), $\sigma_{P}^{2}$ = phenotypic variance among families or populations represented by F_1_ hybrids from each testcross combination of half-siblings, and h^2^ = narrow-sense heritability) [118,121]. As there were no diallel hybridizations, the general combining ability (GCA) and specific combining ability (SCA) were assessed on parent groups for tree vigor and fruit ripening time in a 3 × 3 full mating scheme after Griffing [122]. The data were subjected to a multivariate statistical analysis, namely a principal component analysis (PCA), and a hierarchical clustering algorithm method–Euclidean similarity index performed using PAST software (PAleontological STatistics (PAST) Version 4.09, Natural History Museum, University of Oslo, Norway) [123].

Based on the percentage of F_1_ hybrids that exhibited the quantitative traits expressed individually at an optimal level for selection, the minimum number of offspring was calculated that would have accumulated and associated the desired quantitative features. The Williams procedure [83] was used, but the statistical calculations were more complex and were performed separately within the direct hybridizations and the five testcrosses and finally for the overall experiment.

## 5. Conclusions

Our results illustrated the complexity of apple breeding using hybridization as well as the opportunities for breeders to achieve breeding goals, which must be associated with the current problems facing humanity. The expansion of the world’s population, the need to protect humanity’s food resources, climate change, and other risk factors for cultivated species are all arguments that must be considered in the future. The apple is one of the most important fruit species, and the reduction in funds allocated to research for the development of varieties and the conservation of genetic resources can dramatically amplify the vulnerability of the species to various stress factors. New hybridizations using parental forms with a different genetic dowry from most of the well-known and widespread varieties in the world could widen the genetic base of the new varieties. In this way, the genetic vulnerability of the cultivated apple and the risk of catastrophes (new races and increased virulence of diseases, pests, climate change, etc.) would decrease.

## Figures and Tables

**Figure 1 plants-12-00903-f001:**
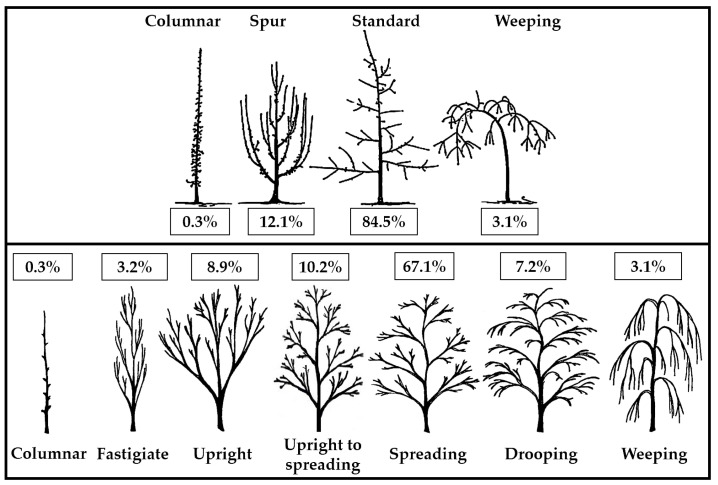
Distribution of F_1_ hybrids (%) in tree architectural ideotypes (tree growth or habitus) according to the classification of Lespinasse et al. (**top**) [40] and UPOV (**bottom**; adapted both from edible and ornamental cultivars) [41,42].

**Figure 2 plants-12-00903-f002:**
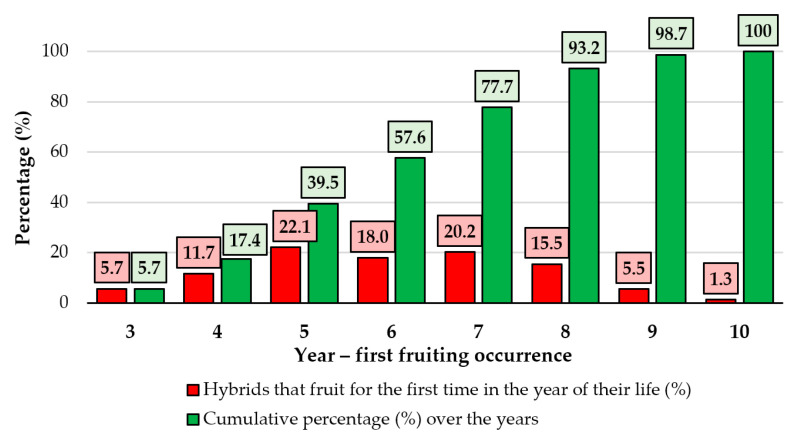
The length of the juvenile period in the F_1_ apple hybrids in the experiment expressed by the year of the first fruiting and the cumulative percentage over the years.

**Figure 3 plants-12-00903-f003:**
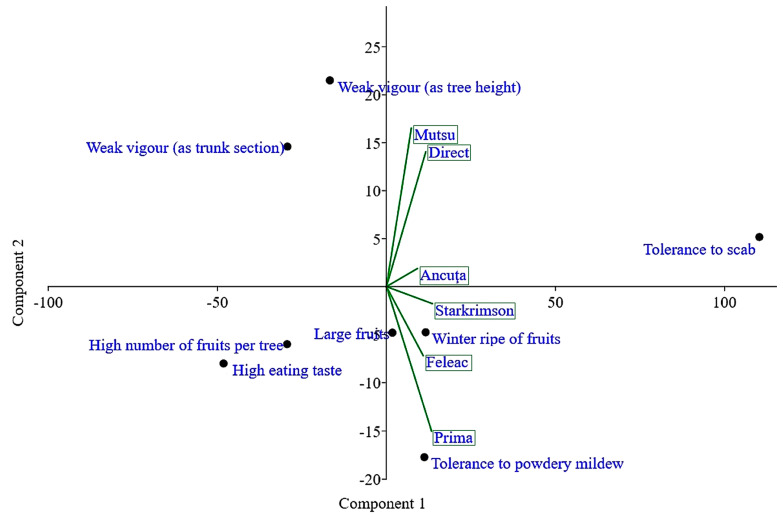
Principal component analysis (PCA) for the types of hybridizations performed (direct hybridizations and five testcrosses) and the traits analyzed in the F_1_ hybrids.

**Figure 4 plants-12-00903-f004:**
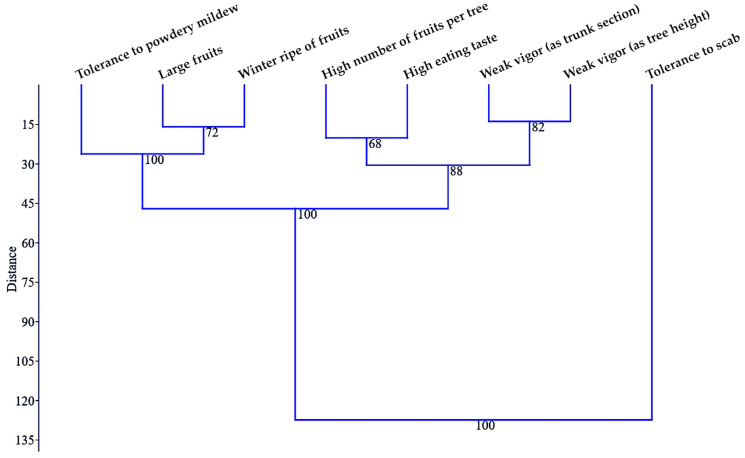
Hierarchical clustering algorithm with paired groups (UPGMA)—Euclidean similarity index of the analyzed traits in F_1_ hybrids belonging to direct hybridizations and five testcrosses.

**Figure 5 plants-12-00903-f005:**
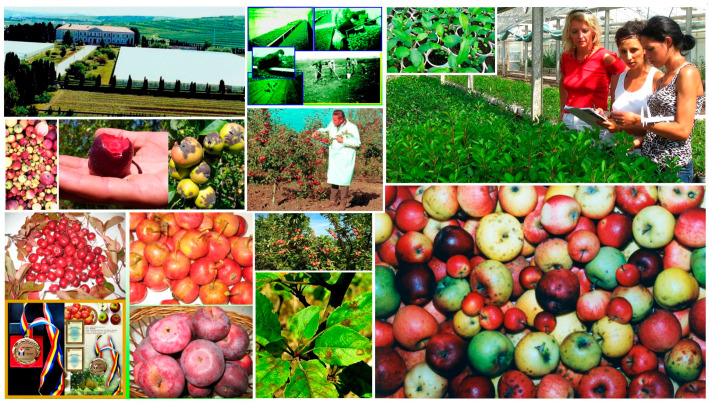
Fruit tree breeding history and activity at HRS Cluj-Napoca: induction of genetic variability (especially through artificial hybridization, selection, and development of new cultivars) and education and training of young specialized scientists.

**Figure 6 plants-12-00903-f006:**
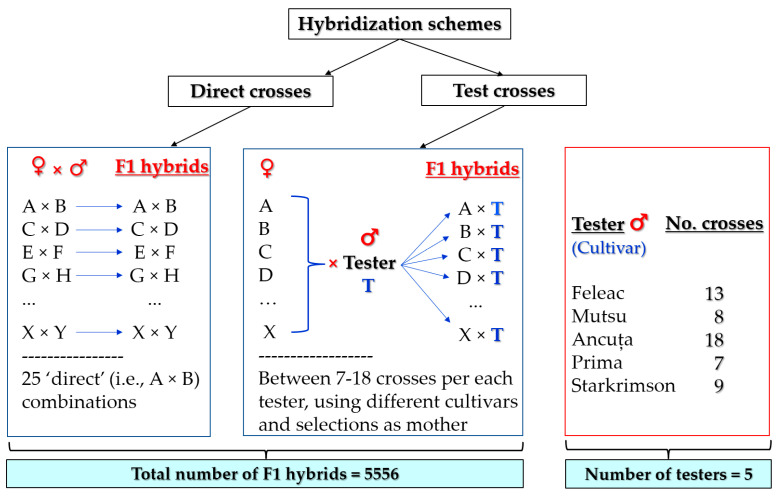
The hybridization schemes performed in the experiment with direct hybridizations and testcrosses.

**Table 1 plants-12-00903-t001:** Trunk cross-sectional area (cm^2^) and tree height (m) in F_1_ apple hybrids in different hybrid combinations.

No.	Hybrid Combination (Parents ♀ × ♂) *	Trunk Cross-Sectional Area (cm^2^)			Height of the Tree (m)		
		Mean ± SD	Sign.	CV%	Mean ± SD	Sign.	CV%
1	Mutsu (s) × Aromat de vară (s)	62.8 ± 6.3	bc	41.5	3.9 ± 0.2	bc	22.4
2	Aromat de vară (s) × Mutsu (s)	53.2 ± 4.7	de	19.9	4.0 ± 0.3	b	17.7
3	Aromat de vară (s) × Reinette Baumann (s)	55.7 ± 12.1	cd	48.9	4.3 ± 0.2	ab	13.3
4	Aromat de vară (s) × Melba (m)	57.3 ± 11.0	c	47.1	3.9 ± 0.4	bc	28.4
5	Aromat de vară (s) × Ancuța (m)	43.9 ± 4.4	e	37.5	3.5 ± 0.1	bcd	18.0
6	Mutsu (s) × Roșu de Cluj (w)	52.6 ± 9.7	de	45.3	3.8 ± 0.3	bcd	23.5
7	Reinette Baumann (s) × Golden spur (w)	44.5 ± 8.0	e	40.4	3.8 ± 0.3	bcd	15.0
	Mean of group (strong vigor ♀)	52.9			3.9		
8	Ardelean (m) × Feleac (s)	65.1 ± 6.8	ab	47.0	5.1 ± 0.2	a	24.9
9	Prima (m) × Feleac (s)	67.7 ± 8.2	a	29.8	3.6 ± 0.1	bcd	10.5
10	Ancuța (m) × Mutsu (s)	62.1 ± 10.9	bc	49.6	3.3 ± 0.3	d	30.8
11	Prima (m) × Ardelean (m)	70.1 ± 6.8	a	44.8	4.8 ± 0.1	a	15.1
12	Ardelean (m) × Prima (m)	71.2 ± 8.8	a	37.5	3.9 ± 0.2	bc	17.3
13	Ardelean (m) × Ancuța (m)	33.2 ± 6.0	f	48.0	3.0 ± 0.2	e	17.0
14	Kinrei (m) × Ancuța (m)	49.1 ± 6.3	e	31.6	4.3 ± 0.2	ab	16.2
15	Golden Delicious (m) × Ancuța (m)	52.7 ± 9.5	de	48.1	4.0 ± 0.2	b	17.7
16	Kinrei (m) × Jonathan (m)	63.9 ± 4.4	b	15.7	3.4 ± 0.2	cd	12.3
17	Ancuța (m) × Roșu de Cluj (w)	65.8 ± 23.5	ab	71.5	4.1 ± 0.4	b	22.9
18	Ancuța (m) × Golden spur (w)	38.9 ± 4.4	f	23.0	3.1 ± 0.1	e	8.0
19	Ancuța (m) × Starkrimson (w)	51.0 ± 8.0	de	41.6	3.4 ± 0.3	cd	24.8
	Mean of group (medium vigor ♀)	57.6			3.8		
20	Roșu de Cluj (w) × Feleac (s)	60.6 ± 6.7	bc	22.2	4.3 ± 0.4	ab	20.4
21	Roșu de Cluj (w) × Ancuța (m)	48.5 ± 8.6	e	37.0	3.3 ± 0.3	d	19.9
22	Roșu de Cluj (w) × Kaltherer Böhmer (m)	52.7 ± 5.2	de	28.0	3.2 ± 0.2	de	18.6
23	Starkrimson (w) × Ancuța (m)	44.3 ± 11.9	e	60.1	3.3 ± 0.3	d	23.0
24	Roșu de Cluj (w) × Golden spur (w)	51.0 ± 9.8	de	54.8	3.2 ± 0.2	de	25.0
25	Roșu de Cluj (w) × Roșu de Cluj (w)	53.4 ± 10.3	de	51.1	3.5 ± 0.2	bcd	16.5
	Mean of group (weak vigor ♀)	51.8			3.5		
	Mean of all combinations	54.9			3.8		

* For each parent, growth vigor is presented as follows: (w)—weak vigor; (m)—medium vigor; (s)—strong vigor. The means on the same column for all hybrid combinations followed by different letters were significantly different according to Duncan’s MRT test (*p* < 0.05).

**Table 2 plants-12-00903-t002:** Classification of F_1_ hybrids from all hybridizations into three vigor classes depending on the trunk cross-sectional area—TCSA (cm^2^) and tree height (m).

Vigor of Parents		Total F_1_	Number of Hybrids and Their Percentage with Vigor Tree					
			Trunk Cross-Sectional Area (cm^2^)			Height of the Tree (m)		
Mother ♀	Father ♂		Weak	Medium	Strong	Weak	Medium	Strong
			(<38.5)	(38.5–78.5)	(>78.5)	(<3.5)	(3.5–4.5)	(>4.5)
Strong	Strong	312	27	183	102	18	135	159
	Medium	318	39	213	66	39	162	117
	Weak	747	108	405	234	90	423	234
Total group (♀ strong)		1377	174	801	402	147	720	510
		(100%)	(12.6%)	(58.2%)	(29.2%)	(10.7%)	(52.3%)	(37.0%)
Medium	Strong	783	118	549	116	93	357	333
	Medium	1503	228	1077	198	249	810	444
	Weak	678	126	492	60	132	357	189
Total group (♀ medium)		2964	472	2118	374	474	1524	966
		(100%)	(15.9%)	(71.5%)	(12.6%)	(16.0%)	(51.4%)	(32.6%)
Weak	Strong	576	123	300	153	105	309	162
	Medium	534	135	315	84	120	330	84
	Weak	105	27	60	18	30	69	6
Total group (♀ weak)		1215	285	675	255	255	708	252
		(100%)	(23.5%)	(55.6%)	(21.0%)	(21.0%)	(58.3%)	(20.7%)
Total general		5556	931	3594	1031	876	2952	1728
		(100%)	(16.8%)	(64.7%)	(18.6%)	(15.8%)	(53.1%)	(31.1%)

For each group of parents, the calculated percentage is shown in parentheses. The classification of hybrids into a particular vigor category based on TCSA or tree height (weak, medium, or strong) was done arbitrarily.

**Table 3 plants-12-00903-t003:** The effects of general combining ability (GCA) and specific combining ability (SCA) on the vigor of F_1_ apple hybrids expressed by the cross-sectional area of the trunk and tree height and depending on the vigor of the parent varieties.

Parents (Vigor)	SCA Effects			GCA Effects	SCA Constancy
	Parents (Vigor)				
	Strong	Medium	Weak		
Trunk Cross-Sectional Area (cm^2^)					
Weak Vigor					
Strong	−0.1863	+0.1897	−0.0034	−0.4405 °°	−0.0805
Medium		−0.1334	−0.0563	−0.0960	−0.0789
Weak			+0.0596	+0.5365 ***	−0.0969
Medium Vigor					
Strong	+0.3447	+0.2183	−0.5630	−0.0812	+0.0462
Medium		−0.5770	+0.3587	+0.5477 **	−0.0479
Weak			+0.2043	−0.4665 °°	+0.0867
Strong Vigor					
Strong	−0.1326	−0.4201	+0.5526 *	+0.5202 **	+0.1311
Medium		+0.7074	−0.2874	−0.4403 °°	+0.0197
Weak			−0.2652	−0.0799	+0.0842
**Height of the Tree (m)**					
Weak Vigor					
Strong	+0.0211	+0.1229	−0.1441	−0.5632 °°	−0.1101
Medium		+0.0488	−0.1711	+0.0304	−0.1056
Weak			+0.3158	+0.5334 **	−0.1028
Medium Vigor					
Strong	−0.0596	+0.1198	−0.0602	−0.5368 °	−0.1865
Medium		−0.1609	+0.0411	−0.0849	−0.1874
Weak			+0.0191	+0.4519 *	−0.1928
Strong Vigor					
Strong	−0.0019	−0.1872	+0.1891	+1.1199 ***	−0.2038
Medium		+0.0614	+0.1258	−0.0963	−0.2138
Weak			−0.3149	−1.0236 °°°	−0.2134

The significance level symbols used for significant positive values: * *p* < 0.05, ** *p* < 0.01, and *** *p* < 0.001, respectively for significant negative values: ° *p* < 0.05, °° *p* < 0.01, and °°° *p* < 0.001. The classification of genotypes into a particular vigor category based on TCSA or tree height (weak, medium, or strong) was done arbitrarily.

**Table 4 plants-12-00903-t004:** Number of fruits per tree and fruit size in F_1_ apple hybrids in different combinations of hybrids according to parents.

Nr.	Hybrid Combination (Parents ♀ × ♂)	Number of Fruits per Tree *			Fruit Size **		
		Mean ± SD	Sign.	CV%	Mean ± SD	Sign.	CV%
1	Mutsu × Aromat de vară	3.8 ± 0.6	cd	60.2	7.5 ± 0.7	b	35.9
2	Mutsu × Roșu de Cluj	2.5 ± 0.8	e	69.3	6.3 ± 0.8	cd	23.8
3	Reinette Baumann × Golden spur	7.0 ± 1.7	a	42.9	5.0 ± 2.0	d	69.3
4	Aromat de vară × Reinette Baumann	2.0 ± 1.0	e	86.6	8.0 ± 1.1	a	21.7
5	Aromat de vară × Melba	5.5 ± 1.7	b	75.2	8.0 ± 0.6	a	19.4
6	Aromat de vară × Ancuța	3.8 ± 0.9	cd	90.1	6.1 ± 0.4	cd	23.1
7	Aromat de vară × Mutsu	4.6 ± 1.7	bc	85.0	8.8 ± 0.7	a	18.7
8	Ardelean × Feleac	4.2 ± 0.8	c	80.1	7.0 ± 0.4	bc	22.1
9	Prima × Feleac	1.5 ± 0.5	e	81.6	7.5 ± 0.5	b	16.3
10	Prima × Ardelean	4.8 ± 0.6	bc	60.4	7.6 ± 0.3	b	16.2
11	Ardelean × Prima	4.4 ± 1.2	c	76.8	8.1 ± 0.5	a	19.2
12	Ardelean × Ancuța	3.4 ± 0.7	d	69.6	7.6 ± 0.4	b	16.6
13	Ancuța × Mutsu	3.1 ± 1.3	d	91.7	7.9 ± 0.5	ab	18.5
14	Kinrei × Ardelean	3.0 ± 1.0	d	57.7	8.0 ± 0.8	a	19.7
15	Golden Delicious × Ancuța	5.0 ± 1.0	bc	34.6	8.0 ± 1.0	a	21.7
16	Roșu de Cluj × Ancuța	5.5 ± 0.9	b	31.5	7.8 ± 0.8	ab	19.2
17	Roșu de Cluj × Kaltherer Böhmer	6.6 ± 1.5	a	61.1	8.3 ± 0.6	a	19.3
18	Kinrei × Jonathan	4.0 ± 1.9	cd	92.6	7.0 ± 0.9	bc	30.3
19	Ancuța × Golden spur	3.0 ± 1.4	d	90.5	7.0 ± 1.0	bc	24.2
20	Ancuța × Starkrimson	3.1 ± 1.1	d	92.1	6.1 ± 0.6	cd	24.0
21	Starkrimson × Ancuța	4.8 ± 1.4	bc	59.9	5.5 ± 1.5	d	54.5
	Mean of all combinations	4.1		70.9	7.3		

* The assessment of the number of fruits per tree was carried out by grading them (regardless of the size of the fruit) as follows: 1—very small number of fruits per tree (under 6–8); 3—small number of fruits per tree (between 9–10 and 20–30); 5—average number of fruits per tree (between 31–40 and 60–70); 7—large number of fruits per tree (between 71–80 and 90–100); 9—very large number of fruits per tree (over 100). ** The assessment of the fruit size was performed by grading them as follows: 1—very small fruits (<50 g); 3—small fruits (50–85 g); 5—medium fruits (85–125 g); 7—large fruits (125–150 g); 9—very large fruits (over 150 g). The means on the same column for all hybrid combinations followed by different letters were significantly different according to Duncan’s MRT test (*p* < 0.05).

**Table 5 plants-12-00903-t005:** Fruit ripening time and fruit taste in F_1_ apple hybrids according to parents.

No.	Hybrid Combination (Parents ♀ × ♂) *	Fruit Ripening Time (Days) **			Fruit Taste ***		
		Mean ± SD	Sign.	CV%	Mean ± SD	Sign.	CV%
1	Aromat de vară (s) × Melba (s)	141.0 ± 14.7	cd	32.9	4.7 ± 0.8	c	44.2
2	Melba (s) × Prima (a)	150.0 ± 3.1	bc	4.5	–	–	–
3	Aromat de vară (s) × Reinette Baumann (w)	152.0 ± 17.4	bc	25.6	4.9 ± 1.2	c	51.6
4	Aromat de vară (s) × Ancuța (w)	148.6 ± 10.7	cd	26.9	5.0 ± 0.5	bc	36.2
5	Aromat de vară (s) × Mutsu (w)	174.0 ± 14.7	a	18.9	5.2 ± 0.8	b	34.4
	Mean of group (summer ♀)	153.1					
6	Ardelean (a) × Clar alb (s)	133.3 ± 8.3	cd	18.8	–	–	–
7	Prima (a) × Ardelean (a)	146.4 ± 3.6	cd	9.1	6.4 ± 0.2	a	19.2
8	Ardelean (a) × Prima (a)	164.4 ± 13.4	b	23.0	6.0 ± 0.3	ab	17.8
9	Ardelean (a) × Feleac (w)	142.7 ± 12.3	cd	28.6	6.5 ± 0.3	a	21.0
10	Ardelean (a) × Ancuța (w)	140.0 ± 16.0	cd	32.6	5.2 ± 0.5	b	32.4
11	Prima (a) × Feleac (w)	165.0 ± 9.8	b	16.8	6.8 ± 0.8	a	26.3
	Mean of group (autumn ♀)	148.6					
12	Mutsu (w) × Aromat de vară (s)	144.3 ± 14.1	cd	25.9	4.5 ± 0.5	cd	45.5
13	Starking (w) × Clar alb (s)	151.8 ± 10.5	bc	22.9	–	–	–
14	Kinrei (w) × Ardelean (a)	151.7 ± 14.2	bc	23.0	3.3 ± 0.8	d	62.0
15	Mutsu (w) × Roșu de Cluj (w)	126.7 ± 18.6	d	35.9	6.0 ± 0.8	ab	27.2
16	Reinette Baumann (w) × Golden spur (w)	174.0 ± 14.7	a	18.9	5.0 ± 1.0	bc	40.0
17	Roșu de Cluj (w) × Feleac (w)	165.0 ± 15.0	b	18.2	6.0 ± 1.6	ab	54.4
18	Ancuța (w) × Mutsu (w)	172.5 ± 10.9	a	18.0	4.9 ± 0.7	c	40.2
19	Ancuța (w) × Roșu de Cluj (w)	155.0 ± 3.2	b	4.4	–	–	–
20	Roșu de Cluj (w) × Ancuța (w)	165.0 ± 15.0	b	18.2	6.0 ± 0.8	ab	38.5
21	Golden Delicious (w) × Ancuța (w)	167.1 ± 11.1	ab	17.5	6.0 ± 0.5	ab	21.1
22	Kinrei (w) × Jonathan (w)	162.0 ± 12.0	b	16.6	6.1 ± 0.4	ab	19.2
23	Ancuța (w) × Golden spur (w)	186.0 ± 14.7	a	17.7	4.6 ± 1.3	c	49.5
24	Ancuța (w) × Starkrimson (w)	167.1 ± 11.0	ab	17.5	5.3 ± 0.7	b	30.6
25	Roșu de Cluj (w) × Kaltherer Böhmer (w)	–	–	–	4.8 ± 0.6	c	39.8
	Mean of group (winter ♀)	160.6					
	Mean of all combinations	156.1			5.4		

* For each parental combination, the ripening time of the fruit is shown for both parents in parentheses as follows: (s)—summer; (a)—autumn; (w)—winter. ** The number of days required from the beginning of flowering to optimal fruit consumption maturity. *** The assessment of the fruit taste was conducted via tasting with notes using the following scale: 1—very poor; 3—poor; 5—medium; 7—good; 9—very good. The means on the same column for all hybrid combinations followed by different letters were significantly different according to Duncan’s MRT test (*p* < 0.05).

**Table 6 plants-12-00903-t006:** Classification of F_1_ hybrids from all hybridizations into three classes depending on the fruit ripening time according to parents and their crosses *.

Fruit Ripening Time		Total F_1_	Hybrids Ripening (Number and %)		
Mother ♀	Father ♂		Summer	Autumn	Winter
Summer	Summer	177	132	36	9
	Autumn	84	33	45	6
	Winter	366	72	207	87
Total group (♀ Summer)		627	237	288	102
		(100%)	(37.8%)	(45.9%)	(16.3%)
Autumn	Summer	168	36	129	3
	Autumn	765	72	453	240
	Winter	672	66	381	225
Total group (♀ Autumn)		1605	174	963	468
		(100%)	(10.8%)	(60.0%)	(29.2%)
Winter	Summer	291	93	156	42
	Autumn	363	27	225	111
	Winter	2598	165	1302	1131
Total groups (♀ Winter)		3252	285	1683	1284
		(100%)	(8.8%)	(51.8%)	(39.5%)
Total general		5484	696	2934	1854
		(100%)	(12.7%)	(53.5%)	(33.8%)

* For each group of parents, the calculated percentage is shown in parentheses.

**Table 7 plants-12-00903-t007:** The effects of general combining ability (GCA) and specific combining ability (SCA) on the ripening time of the fruit of F_1_ apple hybrids depending on the ripening time of the parents.

Parents (Fruit Ripening)	SCA Effects			GCA Effects	SCA Constancy
	Parents (Fruit Ripening)				
	Summer	Autumn	Winter		
Summer Ripening					
Summer	+1.1043	−0.4847 °	−0.6197 °	+1.7457 ***	+0.2205
Autumn		+0.2393	+0.2453	−0.7488 °°°	+0.0586
Winter			+0.3743	−0.9969 °°°	+0.1331
Autumn Ripening					
Summer	−1.8263 °°°	+1.1194 **	+0.7070	−0.7752 °°°	+0.6783
Autumn		−0.9340	−0.1853	+0.8796 ***	+0.4455
Winter			−0.5216	−0.1045	+0.0689
Winter Ripening					
Summer	+0.7263	−0.6207	−0.1057	−0.9521 °°°	+0.0117
Autumn		+0.6673	−0.0467	−0.1387	+0.0072
Winter			+0.1523	+1.0908 ***	−0.1798

Significance level symbols used for significant positive values: * *p* < 0.05, ** *p* < 0.01, and *** *p* < 0.001; and for significant negative values: ° *p* < 0.05, °° *p* < 0.01, and °°° *p* < 0.001.

**Table 8 plants-12-00903-t008:** Average scores of F_1_ apple hybrids from 22 hybrid combinations for apple scab (*Venturia inaequalis*) and powdery mildew (*Podosphaera leucotricha*) attack degree (AD%) *.

No.	Hybrid Combination (Parents ♀ × ♂)	Apple Scab			Powdery Mildew		
		Mean ± SD	Sign.	CV%	Mean ± SD	Sign.	CV%
1	Mutsu × Aromat de vară	1.68 ± 0.13	bc	31.2	3.06 ± 0.32	b	45.2
2	Mutsu × Roșu de Cluj	2.23 ± 0.12	a	13.2	2.70 ± 0.53	c	47.9
3	Reinette Baumann × Golden spur	1.44 ± 0.12	cd	18.1	2.44 ± 0.57	cd	52.6
4	Aromat de vară × Reinette Baumann	1.40 ± 0.13	cd	20.2	3.08 ± 0.23	b	19.3
5	Aromat de vară × Melba	1.87 ± 0.17	b	22.1	3.97 ± 0.33	a	20.5
6	Aromat de vară × Ancuța	1.77 ± 0.13	b	27.6	2.20 ± 0.17	cd	28.3
7	Aromat de vară × Mutsu	1.35 ± 0.17	cd	25.3	1.90 ± 0.17	d	18.2
8	Ardelean × Feleac	1.48 ± 0.09	cd	27.6	2.95 ± 0.19	bc	31.0
9	Prima × Feleac	1.28 ± 0.13	d	57.4	3.08 ± 0.57	b	45.8
10	Prima × Ardelean	1.44 ± 0.11	cd	34.3	3.59 ± 0.31	b	39.0
11	Ardelean × Prima	1.74 ± 0.11	b	18.6	3.43 ± 0.60	b	52.8
12	Ardelean × Ancuța	1.49 ± 0.14	cd	31.7	3.81 ± 0.39	ab	35.5
13	Roșu de Cluj × Feleac	1.35 ± 0.17	cd	25.3	3.65 ± 0.25	b	13.7
14	Ancuța × Mutsu	1.20 ± 0.08	d	19.9	2.56 ± 0.37	c	41.4
15	Kinrei × Ardelean	2.17 ± 0.28	a	31.6	3.17 ± 0.31	b	24.4
16	Ancuța × Roșu de Cluj	1.25 ± 0.10	d	15.3	3.80 ± 0.64	ab	33.6
17	Golden Delicious × Ancuța	1.76 ± 0.09	b	13.9	3.21 ± 0.37	b	30.9
18	Roșu de Cluj × Ancuța	1.60 ± 0.14	bc	17.7	4.80 ± 0.24	a	10.2
19	Roșu de Cluj × Kaltherer Böhmer	1.68 ± 0.15	bc	25.4	2.85 ± 0.33	c	33.1
20	Kinrei × Jonathan	1.72 ± 0.10	b	13.3	2.60 ± 0.38	c	33.1
21	Ancuța × Golden spur	1.60 ± 0.08	bc	10.2	3.70 ± 0.33	b	17.9
22	Ancuța × Starkrimson	1.94 ± 0.13	b	18.6	2.80 ± 0.15	c	14.3
	Mean of all combinations	1.6			3.2		

* The assessment of the apple scab and powdery mildew attack was carried out by grading the varieties as follows: 1—without attack (attack degree (AD%) = 0); 2—very weak attack (AD% = 0.1–1.0); 3—weak attack (AD% = 1.1–5.0); 4—medium attack (AD% = 5.1–15.0); 5—strong attack (AD% = 15.1–20.0); 6—very strong attack (AD% > 20.1). The means on the same column for all hybrid combinations followed by different letters were significantly different according to Duncan’s MRT test (*p* < 0.05).

**Table 9 plants-12-00903-t009:** Genetic parameters for traits analyzed in testcross combinations depending on tester.

Tester	Mean Value F_1_ (Half-Sib Families)	GCV (%)	H^2^	h^2^	Expected Response to Selection (R)	
					Absolute	Relative
Trunk cross-sectional area—TCSA (cm^2^)						
Feleac	62.12	24.7	0.861	0.312	16.13	25.98
Mutsu	45.83	19.5	0.688	0.266	11.56	25.22
Ancuta	53.69	25.3	0.828	0.282	14.49	27.00
Prima	58.75	14.8	0.803	0.261	9.09	15.47
Starkrimson	65.23	27.5	0.882	0.408	21.44	32.86
**Height of the Tree (m)**						
Feleac	4.61	10.0	0.779	0.241	0.49	10.63
Mutsu	3.90	14.9	0.794	0.416	0.84	21.54
Ancuta	4.17	13.2	0.864	0.312	0.58	13.91
Prima	4.40	5.0	0.651	0.130	0.20	4.55
Starkrimson	4.42	13.9	0.864	0.389	0.74	16.74
**Number of Fruits per Tree (marks; scale 1–3–5–7–9)**						
Feleac	4.03	28.3	0.672	0.178	1.2	29.83
Mutsu	4.10	–	–	–	–	–
Ancuta	4.25	16.5	0.587	0.082	0.55	12.83
Prima	4.14	–	–	–	–	–
Starkrimson	4.06	29.1	0.698	0.212	1.32	32.6
**Fruit Size (marks; scale 1–3–5–7–9)**						
Feleac	7.87	2.6	0.531	0.032	0.11	1.34
Mutsu	7.00	14.3	0.683	0.280	1.34	19.17
Ancuta	7.51	6.0	0.599	0.094	0.37	4.91
Prima	6.20	41.8	0.946	0.384	2.35	37.93
Starkrimson	7.39	11.5	0.760	0.278	1.01	13.67
**Ripening Time of the Fruit (days)**						
Feleac	169.3	6.9	0.684	0.183	12.41	7.33
Mutsu	158.6	11.7	0.743	0.373	27.26	17.19
Ancuta	167.6	7.7	0.716	0.202	13.84	8.26
Prima	164.7	8.3	0.779	0.241	14.26	8.66
Starkrimson	172.1	12.6	0.900	0.427	25.63	14.89
**Taste of the Fruit (marks; scale 1–3–5–7–9)**						
Feleac	5.81	–	–	–	–	–
Mutsu	5.10	8.1	0.566	0.101	0.37	7.21
Ancuta	5.52	8.3	0.628	0.117	0.41	7.34
Prima	4.57	49.8	0.975	0.410	7.91	41.90
Starkrimson	4.98	21.4	0.829	0.351	1.29	25.93
**Response to Apple Scab Attack (marks; scale 1–2–3–4–5–6)**						
Feleac	1.64	12.5	0.677	0.153	0.20	12.02
Mutsu	1.70	16.6	0.668	0.245	0.36	21.08
Ancuta	1.71	17.3	0.876	0.309	0.30	17.54
Prima	1.37	(22.3)	–	–	–	–
Starkrimson	1.65	14.9	0.819	0.341	0.30	18.09
**Response to Powdery Mildew Attack (marks; scale 1–2–3–4–5–6)**						
Feleac	2.58	18.2	0.763	0.228	0.49	19.16
Mutsu	2.50	9.8	0.554	0.079	0.19	7.74
Ancuta	2.59	26.9	0.825	0.339	0.81	31.45
Prima	1.85	41.0	0.891	0.337	0.75	40.70
Starkrimson	2.59	22.7	0.852	0.376	0.71	25.73

Note: GCV: genotypic coefficient of variation; H^2^: heritability in broad sense; h^2^: heritability in narrow sense.

**Table 10 plants-12-00903-t010:** The percentage of F_1_ apple hybrids that showed the analyzed traits expressed at an optimal level for selection.

Traits	Type of Hybridization					
	Direct	Testcross Using as Tester:				
		Feleac	Mutsu	Ancuţa	Prima	Starkrimson
High eating taste	4.6	9.2	4.8	3.9	4.0	1.4
Large fruits	22.5	32.7	21.4	30.5	22.7	22.9
Tolerance to scab	76.6	66.9	57.8	59.3	76.0	77.1
Tolerance to powdery mildew	17.9	25.0	22.2	23.7	45.4	25.7
High number of fruits per tree	10.6	13.5	14.3	13.6	14.7	8.6
Winter ripe of fruits	20.6	32.0	26.2	34.6	25.3	37.1
Weak vigor (as trunk section, TCSA)	16.5	10.5	30.4	18.8	2.7	5.7
Weak vigor (as tree height)	27.0	4.8	32.6	15.9	4.0	11.5

**Table 11 plants-12-00903-t011:** The minimum number of F_1_ apple hybrids from hybridizations required to obtain an individual that inherited the optimal level for quantitative traits desired in selection.

Type of Hybridizations		Minimum Number of Offspring Required to Obtain an Individual Hybrid with the Following Associated Quantitative Traits *:					
		a + b	a + b + c + d	a + b+ c + d + e	a + b + c + d + e + f	a + b + c + d + e + f + g1	a + b + c + d + e + f + g2
Direct—type A × B (21 combinations)		96	704	6647	32,270	195,578	119,520
Testcross with paternal tester:	Feleac	33	198	1472	4600	43,815	95,845
	Mutsu	97	758	5305	20,250	66,612	62,116
	Ancuţa	84	598	4398	12,712	67,617	79,950
	Prima	110	319	2171	8582	317,865	214,559
	Starkrimson	311	1574	18,304	49,337	865,569	429,021
Average		122	692	6383	21,292	259,509	166,835

* a—High eating taste; b—large fruits; c—tolerance to apple scab attack; d—tolerance to powdery mildew attack; e—high number of fruits per tree; f—winter ripening of fruits; g1—weak vigor (as trunk section area); g2—weak vigor (as tree height).

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
