# Peer review of "Quantitative Traits of Interest in Apple Breeding and Their Implications for Selection"

_plants, 2023, doi:10.3390/plants12040903_

Round 1

Reviewer 1 Report

The manuscript is a very nice piece of work. Dealing with trees is not an easy task and not short in time. Thus I think this paper presents results after several years of work. It shows as well interesting genetic estimations as GCA and SCA or heritabilities,...Those estimations are valuable for developing new apple varieties not only with the varieties employed in the study but with other germplasm.

Some minor remarks:

Line 13. Please refer always F1 with subindex F1

Line 20 remove ‘but associated’ as  they cannot be associated if they are independent traits (genetically speaking). Probably authors meant that both characters need to be present in an new variety, if so please re-phrase.

Table 11. There are not references of how the data o this table have been calculated. I haven't found this neither in the statistical analysis section. Please complete information about this. 

Author Response

Dear Reviewer,

We appreciate you taking the time to read and appreciate our manuscript, which we believe will be of interest to Plants journal readers.

According on your suggestions, the following revisions were made to the article:

Issue 1

Line 13. Please refer always F1 with subindex F1

Response I_1: Thank you, we concur that it is common to mark the first hybrid generation with the subscript/subindex F1. We updated the entire manuscript with the necessary adjustments.

Issue 2

Line 20 remove ‘but associated’ as they cannot be associated if they are independent traits (genetically speaking). Probably authors meant that both characters need to be present in an new variety, if so please re-phrase.

Response I_2: Thank you, we appreciate your insightful remark, and we have made the required changes to eliminate any ambiguity. Consequently, the revised wording from the relevant line/lines is as follows:

"For two independent traits essential in selection (fruit size and quality), but incorporated together in the same hybrid, the statistical number was between about 30 and 300". 

Issue 3

Table 11. There are not references of how the data o this table have been calculated. I haven't found this neither in the statistical analysis section. Please complete information about this. 

Response I_3: Thank you for your remark; it appears that we overlooked the necessary explanations regarding the calculation of the minimum statistical number of hybrids required to produce offspring with the desired phenotypic traits. Based on your helpful observation, we have made the necessary adjustments so that readers can understand our calculation procedures and, as a result, the findings and their significance.

Thus, at the end of subchapter 4.4. (Statistical analysis), we added the following explanatory text:

"Based on the percentage of F1 hybrids that exhibited the quantitative traits ex-pressed individually, at an optimal level for selection, the minimum number of offspring was calculated that would have accumulated and associated the desired quantitative features. The Williams procedure was used [83], but the statistical calculations were more complex, being performed separately within the direct hybridizations and the five testcrosses, and finally for the overall experiment".

Thank you for your helpful suggestions; we hope that the revised manuscript is now suitable for publication and will be a valuable contribution to Plants.

Reviewer 2 Report

The manuscript titled "Quantitative traits in apple breeding and their implication for selection" is a perfectly planned, executed and presented study. Only suggestions for authors is to revise the manuscript for numerous grammatical errors before resubmission. 

Author Response

Dear Reviewer,

We are grateful that you have taken the time to read and appreciate our paper, which we think the readers of the Plants journal will find to be of interest.
In accordance with the advice that you provided, the manuscript underwent extensive revisions, including one that was carried out by a native expert in plant breeding.

Thank you.